# SD4Match: Learning to Prompt Stable Diffusion Model for Semantic Matching

## Abstract

In this paper, we address the challenge of matching semantically similar keypoints across image pairs. Existing research indicates that the intermediate output of the UNet within the Stable Diffusion (SD) framework can serve as robust image feature maps for such a matching task. We demonstrate that by employing a basic prompt tuning technique, the inherent potential of Stable Diffusion can be harnessed, resulting in a significant enhancement in accuracy over previous approaches. We further introduce a novel conditional prompting module that conditions the prompt on the local details of the input image pairs, leading to a further improvement in performance. We designate our approach as SD4Match, short for Stable Diffusion for Semantic Matching. Comprehensive evaluations of SD4Match on the PF-Pascal, PF-Willow, and SPair-71k datasets show that it sets new benchmarks in accuracy across all these datasets. Particularly, SD4Match outperforms the previous state-of-the-art by a margin of 12 percentage points on the challenging SPair-71k dataset.

## 1 Introduction

Matching keypoints between two semantically similar objects is one of the challenges in computer vision. The difficulties arise from the fact that semantic correspondences may exhibit significant visual dissimilarity due to occlusion, different viewpoints, and intra-category appearance differences. Although significant progress has been made (Han et al., 2017; Rocco et al., 2018; Cho et al., 2021; Min et al., 2019a; Li et al., 2023), the problem is far from being completely solved. Recently, Stable Diffusion (SD) (Rombach et al., 2022) has demonstrated a remarkable ability to generate high-quality images based on input textual prompts. Looking specifically at semantic matching, follow-up studies (Tang et al., 2023; Zhang et al., 2023) have further revealed that SD is not only proficient in generative tasks but also applicable to feature extraction. Experiments demonstrate that the SD can perform on par with methods especially designed for semantic matching, paving a new direction in this field. This brings up a yet unanswered question: Have we fully explored the capacity of the SD in matching? Or, how should we harness the information gathered from billions of images stored within the SD to further improve its performance?

Engineering the textual prompt has already been extensively utilized in numerous computer vision tasks, including image generation using Stable Diffusion. In these applications, prompts are meticulously handcrafted to achieve the desired output. Prompt tuning, or direct optimization of prompt embedding, has also been utilized to adapt pre-trained vision-language models, such as CLIP (Radford et al., 2021), to new data domains in tasks like image classification, especially when faced with limited data resources (Zhou et al., 2022b; Khattak et al., 2023). Inspired by the latter strategy, and given that the accuracy of matching is quantifiable and limiting the prompt to the textual domain is unnecessary, we have chosen to directly optimize the prompt on the latent space for semantic matching. In spite of its straightforward nature, we find that learning a single, universal prompt applicable to all images is already highly effective in adapting the SD to semantic matching, and not only improves previous SD-based semantic matchers (Tang et al., 2023; Zhang et al., 2023) but also leads to the state-of-the-art performance over all types of methods.

Current works that explore SD for semantic tasks (matching in (Tang et al., 2023) and segmentation in (Xu et al., 2023)) mimic the textual prompt input of standard image generation SD, by either handcrafting a text input (Tang et al., 2023) or by using an implicit captioner (Xu et al., 2023).

Novel to our work, we find that the choice for prompt, text or otherwise, particularly when including prior information, significantly influences the overall performance. We then introduce two additional prompt tuning schemes tailored specifically for semantic matching: one that leverages explicit prior semantic knowledge and learns a distinct prompt for each object category, and a novel conditional prompting module (CPM) that conditions the prompt on the local feature patches of both images in the image pair to be matched, instead of global descriptors of each individual image. Experiments show that these designs lead to further improvements in matching accuracy.

We can delineate our contributions in this paper as follows:

1. We demonstrate that the performance of Stable Diffusion in the semantic matching task can be significantly enhanced using a straightforward prompt tuning technique.

2. We propose a novel conditional prompting module, which uses the local features of the image pair. Our experiments show that this design supersedes earlier models reliant on the global descriptor of individual images, leading to a noticeable improvement in matching accuracy.

3. We evaluate our approach on the PF-Pascal, PF-Willow, and SPair-71k datasets, establishing new accuracy benchmarks for each. Notably, we achieve an increase of 12 percentage points on the challenging SPair-71k dataset, surpassing the previous state-of-the-art.

## 2 RELATED WORK

**Semantic Correspondence** Early attempts at semantic matching focused on handcrafted features such as HOG (Dalal & Triggs, 2005), SIFT (Lowe, 2004), and SIFT Flow (Liu et al., 2010). SCNet (Han et al., 2017) was the first deep learning method to tackle this problem. Various network architectures have been proposed, addressing the problem from different perspectives, such as metric learning (Choy et al., 2016), neighbourhood consensus (Rocco et al., 2018; Li et al., 2020; Min & Cho, 2021), multilayer feature assembly (Min et al., 2019a; 2020), and transformer-based architecture (Cho et al., 2021; Kim et al., 2022b), etc. Another line of work in this field is learning matching from image-level annotations. SFNet (Lee et al., 2019) uses segmentation masks of images; PMD (Li et al., 2021) employs teacher-student models and learns from synthetic data, while PWarpC (Truong et al., 2022) relies on the probabilistic consistency flow from augmented images. Although significant progress has been made, the majority of these methods are based on ResNet (He et al., 2016), which has inferior representation capability compared to later ViT-based feature extractors like DINO (Caron et al., 2021; Oquab et al., 2023) or iBOT (Zhou et al., 2021), thus limiting their performance. SimSC Li et al. (2023) demonstrates an improvement of 20% in matching accuracy by switching from ResNet to iBOT in its finetuning pipeline.

**Diffusion Model** The pioneering work that formulated image generation as a diffusion process is DDPM (Ho et al., 2020). Since then, numerous follow-up works have been proposed to improve the generation process. DDIM (Song et al., 2020) and PNDM (Liu et al., 2022) accelerate the generation process through the development of new noise schedulers. The works by Dhariwal & Nichol (2021) and Ho & Salimans (2022) enhance the fidelity of the generation by adjusting the denoising step. Another milestone in this field is Stable Diffusion (Rombach et al., 2022), which significantly increases the resolution of generated image by working on the latent space instead of pixel level, paving the way for novel methods in image editing (Brooks et al., 2023; Couairon et al., 2022) and object-oriented image generation (Gal et al., 2022; Ruiz et al., 2023), etc. More recently, Zhao et al. (2023) found that pre-trained Stable Diffusion can also act as a feature extractor, drawing features from images for visual perception tasks. This insight led to studies like DIFT (Tang et al., 2023) and SD+DINO (Zhang et al., 2023), which delve into the impact of timestep and layer on pre-trained SD's capabilities in semantic matching. Our work is closely related to these efforts, but we instead explore how the prompt can be optimized within an SD framework to improve its performance on semantic matching.

**Prompt Tuning** Prompt tuning has gained popularity due to its success in adapting pretrained language models to downstream tasks in natural language processing (Shin et al., 2020; Jiang et al., 2020). COOP (Zhou et al., 2022b) was the first work to introduce prompt tuning to computer vision, adapting CLIP to different data distributions in a few-shot setting. Its successor, COCOOP (Zhou

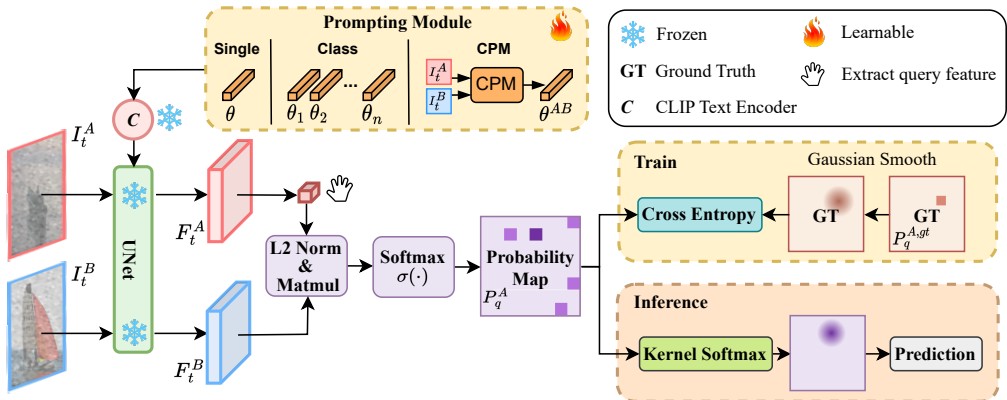

Figure 1: The general pipeline of SD4Match. We present three prompting options for the Stable Diffusion on semantic matching: Single, Class, and conditional prompting module (CPM). The prompt is tuned by the cross-entropy loss between the predicted probability map and the ground-truth probability map of given query points. During inference, we use Kernel-Softmax proposed by Lee et al. (2019) to localize correspondences.

et al., 2022a), conditioned the prompt on input images to enhance generalizability. These have inspired several other prompting methods (Khattak et al., 2023; Zhu et al., 2022). A parallel line of research explores visual prompts, often in the form of masks overlaid on images, to achieve similar objectives (Bahng et al., 2022; Oh et al., 2023). However, most literature has solely focused on prompt tuning for image classification tasks. To the best of our knowledge, our study is the first to apply prompt tuning to the SD model for semantic matching.

## 3 METHOD

In this section, we introduce our method, namely Stable Diffusion for Semantic Matching (SD4Match). We illustrate the general pipeline in Fig. 1. The UNet component within the SD extracts feature maps from input images based on the prompt generated by our prompting module. We present three options for the prompt: one single universal prompt (SD4Match-Single), one prompt per object category (SD4Match-Class), and the conditional prompting module (SD4Match-CPM). The prompt is tuned by the cross-entropy loss between the predicted matching probability and the ground-truth probability of given query points. All modules are kept frozen except for the prompting module during the tuning.

We organize this section as follows: In Sec. 3.1, we briefly introduce the diffusion model and its application as a feature extractor. In Sec. 3.2, we investigate the effect of the existing prompt schemes and introduce our prompt tuning in detail. Finally, in Sec. 3.3, we elaborate on the design of the conditional prompting module and the reasoning behind it.

### 3.1 PRELIMINARY

**Diffuion Model**  The image diffusion model, as proposed in Ho et al. (2020); Song et al. (2020), is a generative model designed to capture the distribution of a set of images. This model consists of two processes: forward and reverse. In the forward process, a clean image $I$ is progressively corrupted by a sequence of Gaussian noise. This corruption follows the equation:

$$I_t = \sqrt{\alpha_t}I_{t-1} + \sqrt{1-\alpha_t}\epsilon_{t-1} \qquad (1)$$

where $\epsilon \sim \mathcal{N}(0, 1)$ and $\alpha_t$ is the coefficient controlling the level of corruption at timestep $t$. When $t = T$ is sufficiently large, image $I_T$ is totally corrupted, resembling a sample of $\mathcal{N}(0, 1)$. In the reverse process, the diffusion model $f_\theta(I_t, t)$ learns to predict the noise $\epsilon_t$ added to the image $I$ at timestep $t$ in the forward process. Therefore, by drawing a sample from $\mathcal{N}(0, 1)$, we can recover its corresponding "original image" by iteratively removing the noise $\epsilon_t$. In Stable Diffusion (Rombach et al., 2022), such a reverse process can condition on various types of input, such as text or other images, to control the content of generated image.

Table 1: Evaluation of existing works on SPair-71k dataset with different prompt types. The results are PCK with $\alpha = 0.1$. The definition of the metric is provided in Sec. 4.2.

| Prompt Type | Empty String | "a photo of a {*object category*}" | Implicit Captioner |
|---|---|---|---|
| DIFT (Tang et al., 2023) | 50.7 | 52.9 | - |
| SD+DINO* (Zhang et al., 2023) | 50.3 | 52.2 | 52.4 |

**Stable Diffusion as Feature Extractor**   The text-to-image Stable Diffusion is found with the capability of extracting semantically meaningful feature maps from images (Zhang et al., 2023; Tang et al., 2023; Luo et al., 2023). Given an input image $I$ and a specific timestep $t$, $I$ is encoded by VAE to the latent representation $z$ which is then corrupted to $z_t$ by Eq. (1). The UNet in Stable Diffusion then predicts the noise at timestep $t$. The resulting feature map is obtained from the output of an intermediate layer of the UNet's decoder during this noise prediction phase. Observations show that the earlier layer of the decoder with a large $t$ captures more abstract and semantic information while the later layer of the decoder with a small $t$ focuses on local texture and details. This is similar to the feature pyramids in ResNet (He et al., 2016). Therefore, careful choices of $t$ and layer are required. For the sake of simplicity, we skip the VAE encoding step and directly refer to the UNet's input as image $I$ rather than latent representation $z$ in the following paragraphs.

## 3.2 PROMPT TUNING FOR SEMANTIC CORRESPONDENCE

We first investigate the impact of various existing prompts on matching accuracy. We evaluated three commonly-used prompts: an empty textual string " " (Zhao et al., 2023); the textual template "a photo of a {*object category*}" which requires the category of the object (Tang et al., 2023), and the implicit captioner borrowed from the image segmentation method (Xu et al., 2023), directly converting the input image into textual embeddings (Zhang et al., 2023). We applied these prompts to two SD-based approaches: DIFT (Tang et al., 2023) and SD+DINO (Zhang et al., 2023). DIFT directly uses the feature map produced by SD2-1 to perform matching, whereas SD+DINO fuses the features from DINOv2 and SD1-5 for better accuracy. To isolate the effect of DINO on matching results, we excluded the DINO feature in SD+DINO, designating this modified setting as SD+DINO*. The results are summarized in Tab. 1. We notice that the SD model keeps the majority of its capability in semantic matching even when supplied with a non-informative empty string. This is attributed to the fact that the timestep $t$ in both works is relatively small ($t = 261$ for DIFT and $t = 100$ for SD+DINO out of the total timestep $T = 1000$) so the information in the input image remains largely intact. Therefore, the image itself is sufficient for most matching cases. With the help of prior knowledge of the object of interest, either in the form of object category or the implicit captioner, the accuracy is improved by about 2 percentage points, reflecting the importance of input-related prompts.

Analogous to the empty string, we search for a single universal prompt that is applied to all images. We randomly initialize the prompt and directly finetune the prompt embeddings with the semantic matching loss proposed by Li et al. (2023). Given two images $I_t^A$ and $I_t^B$ corrupted to timestep $t$, the UNet $f(\cdot)$ of the Stable Diffusion extracts their corresponding feature maps $F_t^A \in \mathbb{R}^{C \times H_A \times W_A}$ and $F_t^B \in \mathbb{R}^{C \times H_B \times W_B}$ by:

$$F_t = f(I_t, t, \theta) \tag{2}$$

where $\theta \in \mathbb{R}^{N \times D}$ is the prompt embedding, $N$ is the prompt length and $D$ is the dimension of the embedding. $F_t^A$ and $F_t^B$ are then L2-normalized along the feature dimension obtaining $\widehat{F}_t^A$ and $\widehat{F}_t^B$. Let $\mathbf{X} = \{(\mathbf{x}_q^A, \mathbf{x}_q^B) \mid q = 1, 2, ..., n\}$ be the ground-truth correspondences provided in the training data. For each query point $\mathbf{x}_q^A = (x_q^A, y_q^A)$ in $I_t^A$, we extract its corresponding feature $\widehat{F}_{t,q}^A \in \mathbb{R}^C$ from $\widehat{F}_t^A$ and compute a correlation map $M_q^A \in [-1, 1]^{H_B \times W_B}$ with the entire $\widehat{F}_t^B$:

$$M_{q,kl}^A = (\widehat{F}_{t,q}^A)^\top \widehat{F}_{t,kl}^B \tag{3}$$

where $\widehat{F}_{t,kl}^B \in \mathbb{R}^C$ are the feature at position $(k, l)$ in $\widehat{F}_t^B$. The correlation map $M_q^A$ is converted to a probability distribution $P_q^A$ by the softmax operation $\sigma(\cdot)$ with temperature $\beta$:

$$\sigma(\mathbf{z})_i = \frac{exp(z_i / \beta)}{\sum_j exp(z_j / \beta)} \tag{4}$$

The loss between $I_t^A$ and $I_t^B$ is the average of the cross-entropy between $P_q^A$ and the ground-truth distribution $P_q^{A,gt}$ of all correspondence pairs $\mathbf{X}$:

$$\mathcal{L} = \frac{1}{n} \sum_{q=1}^{n} H(P_q^{A,gt}, P_q^A) \tag{5}$$

where $P_q^{A,gt}$ is the Dirac delta distribution $\delta(\mathbf{x}_q^B)$. Following Li et al. (2023), we apply a $k \times k$ Gaussian kernel to $P_q^{A,gt}$ for label smoothing. During inference, we use Kernel-Softmax (Lee et al., 2019) to localize the prediction. The entire UNet $f$ is fixed and the only parameter required to update is the prompt embedding $\theta$ during tuning. We refer to this option as SD4Match-Single.

Just like the textual template, we can also learn one prompt for one specific category. Assume we have $n$ classes and a set of $n$ prompt embeddings $\{\theta_1, \theta_2, ..., \theta_n\}$, then Eq. (2) becomes:

$$F_t = f(I_t, t, \Theta(c(I_t))) \tag{6}$$

where $c(I_t) \in \{1, 2, ..., n\}$ is the category of object of interest in $I_t$ and $\Theta(i) = \theta_i$. We denote this as SD4Match-Class.

## 3.3 CONDITIONAL PROMPTING MODULE

An alternative approach to SD4Match-Class involves conditioning the prompt on input images, thus eliminating the need for manual inspection of the object's category. Previous studies have delved into conditional prompts for tasks like image classification (Zhou et al., 2022a; Oh et al., 2023) and image segmentation Xu et al. (2023). In this context, the prompt is conditioned on the global descriptor of the image, typically extracted by ViT-based (Dosovitskiy et al., 2020) feature extractors such as DINOv2 (Oquab et al., 2023) or CLIP (Radford et al., 2021). This descriptor is then projected to match the dimension of the prompt embedding and forwarded to the text encoder accompanied by a learnable positional embedding. While this approach has shown effectiveness for the tasks mentioned above, it might not be the optimal design for finding semantic correspondence. Our reasons are:

1. Semantic matching involves a pair of images. The prompt for this specific pair should be one prompt conditioned on and applied to both images, rather than two different prompts conditioned on each individual image and applied to them separately.

2. Semantic matching relies on the local details of images. The prompt should be conditioned on the local features rather than the global descriptors of the images.

3. The prompt should incorporate a universal head that is applicable to all images. An analogy for this is the prefix "a photo of a" in the textual template.

We therefore propose a novel conditional prompting module (CPM) and illustrate its architecture in Fig. 2. It mainly consists of four modules: a DINOv2 feature extractor, two linear layers $g_d(\cdot)$ and $g_n(\cdot)$ and an adaptive MaxPooling layer $p(\cdot)$. Given a pair of clean images $I^A$ and $I^B$, we first use DINOv2 to extract their local feature patches $\mathcal{F}^A$ and $\mathcal{F}^B$, where $\mathcal{F}^A, \mathcal{F}^B \in \mathbb{R}^{N_{dino} \times D_{dino}}$. We then fuse the local features of two images by concatenating them along the feature dimension and projecting the concatenated feature $\mathcal{F}^{AB} \in \mathbb{R}^{N_{dino} \times 2D_{dino}}$ to the dimension of the prompt embedding $D$ by the linear layer $g_d(\cdot)$, resulting in $\widetilde{\mathcal{F}}^{AB} \in \mathbb{R}^{N_{dino} \times D}$. These operations only explore the feature-wise relationship within the image pair but do not extract the inter-patches information from it. Therefore, we further process $\widetilde{\mathcal{F}}^{AB}$ by another linear layer $g_n(\cdot)$ along the *patch* dimension. This allows information exchanges between different local feature patches, enhancing the capability of the prompt. The output of $g_n(\cdot)$ goes through the adaptive MaxPooling layer $p(\cdot)$ to reduce its patch dimension to $N_{cond}$ so that the prompt will not exceed the maximum prompt length of the SD model, producing $\widehat{\mathcal{F}}^{AB} \in \mathbb{R}^{N_{cond} \times D}$. We then follow the design in Xu et al. (2023), generating the conditional prompt $\theta_{cond}^{AB}$ by $\theta_{cond}^{AB} = \widehat{\mathcal{F}}^{AB} * \Omega_\alpha + \Omega_{pos}$, where $\Omega_\alpha \in \mathbb{R}^{N_{cond} \times D}$ is a conditioning weight and $\Omega_{pos} \in \mathbb{R}^{N_{cond} \times D}$ is a positional embedding. The conditional prompt $\theta_{cond}^{AB}$ is appended after a global prompt $\theta_{global} \in \mathbb{R}^{N_{global} \times D}$, producing the final prompt $\theta^{AB} \in \mathbb{R}^{N \times D}$ and $N = N_{global} + N_{cond}$. Eq. (2) subsequently becomes:

$$F_t^A = f(I_t^A, t, \theta^{AB}), \quad F_t^B = f(I_t^B, t, \theta^{AB}) \tag{7}$$

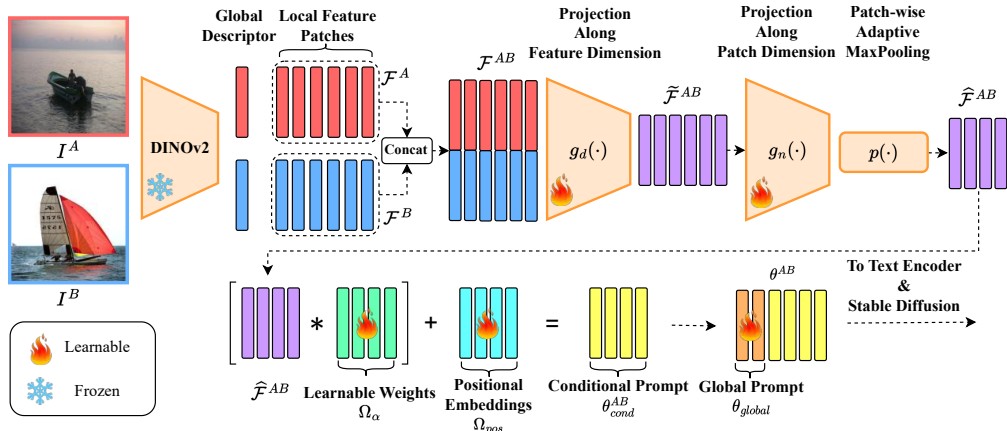

Figure 2: Illustration of the architecture of our conditional prompting module.

Note that the feature map $F_t^A$ changes if its pairing image $I_t^B$ also changes since the prompt is conditioned on both images in the image pair. A benefit of this design is that if multiple objects are present in images, the prompt would focus on the common object between the image pair.

## 4 EXPERIMENTS

In this section, we first provide the implementation details of our method in Sec. 4.1 and introduce datasets and evaluation metrics in Sec. 4.2. We then present the evaluation results and ablation studies in Sec. 4.3 and Sec. 4.4 respectively.

### 4.1 IMPLEMENTATION DETAILS

Our method is implemented in Python using the Huggingface (Gugger et al., 2022; Wolf et al., 2020; von Platen et al., 2022) and PyTorch (Paszke et al., 2019) libraries. We follow DIFT and adopt Stable Diffusion 2-1 as the diffusion model, where the total timestep $T$ is 1000. We utilize the output from the $2^{\text{nd}}$ up block of the UNet as the feature map, and set the timestep $t = 261$ during training. We choose DINOv2-ViT-B/14 (Oquab et al., 2023) as the feature extractor in CPM. The maximum prompt length for Stable Diffusion is 77, which includes two special tokens: SOS and EOS. Therefore, we set the prompt length $N = 75$ in SD4Match-Single and SD4Match-Class, and $N_{global} = 25$ and $N_{cond} = 50$ in SD4Match-CPM, occupying all 77 token positions. For inference, we set the timestep $t = 50$ as empirical tests suggest it provides optimal results, even though our method was trained at $t = 261$. The temperature $\beta$ and the Gaussian smooth kernel $k$ are set to 0.04 and 7, respectively, during training. We train our method using the Adam optimizer (Kingma & Ba, 2014) with a batch size of 9 for 30,000 steps across all experiments. The learning rate is set at $1 \times 10^{-2}$ in all configurations, except for the two linear layers $g_d(\cdot)$ and $g_n(\cdot)$ in the CPM, where it's $1 \times 10^{-3}$. This rate remains constant throughout training. Images are resized to $768 \times 768$ for both the training and testing phases. We train SD4Match on three Quadro RTX 6000 GPUs.

### 4.2 DATASETS AND EVALUATION METRICS

We evaluate our method on three popular semantic correspondence datasets: PF-Pascal (Ham et al., 2016), PF-Willow (Ham et al., 2017), and SPair-71k (Min et al., 2019b). PF-Pascal consists of 2941 training image pairs, 308 validation pairs, and 299 testing pairs spanning across 20 categories of objects. PF-Willow is the supplement to PF-Pascal with 900 testing pairs only. SPair-71K is a larger and more challenging dataset with $53,340$ training pairs, $5,384$ validation pairs, and $12,234$ testing pairs across 18 categories of objects with large scale and appearance variation. Each of the three datasets has non-uniform numbers of ground-truth correspondences.

We follow the common practice in the literature and use the Percentage of Corrected Keypoints (PCK) as the evaluation metric. Given an image pair $(I^A, I^B)$ and its associated correspondence set

Table 2: Evaluation on the SPair-71k dataset at $\alpha = 0.1$. Methods are classified into three categories based on their degree of supervision: (1) methods which are zero-shot and not tuned on the training data of Spair-71k, marked as **Z**. (2) methods using image-level annotations, marked as **I**. (3) methods using ground-truth keypoint annotations, marked as **K**. Best results in each category are **bolded**. Overall, Our method achieves the best results in all of 18 categories and we outperform the second-best method SimSC-iBOT (Li et al., 2023) by 12 percentage points.

| Method | Aero | Bike | Bird | Boat | Bottle | Bus | Car | Cat | Chair | Cow | Dog | Horse | Motor | Person | Plant | Sheep | Train | TV | All |
|---|---|---|---|---|---|---|---|---|---|---|---|---|---|---|---|---|---|---|---|
| **Z** DINO (Caron et al., 2021) | 37.3 | 23.8 | 63.0 | 19.9 | 41.7 | 29.9 | 24.1 | 64.4 | 21.3 | 48.7 | 42.1 | 30.3 | 23.3 | 41.0 | 28.6 | 29.8 | 40.7 | 37.1 | 35.9 |
| DINOv2 (Oquab et al., 2023) | 69.9 | 58.9 | 86.8 | 36.9 | 43.4 | 42.6 | 39.3 | 70.2 | 37.5 | 69.0 | 63.7 | 68.9 | 55.1 | 65.0 | 33.3 | 57.8 | 51.2 | 31.2 | 53.9 |
| DIFT (Tang et al., 2023) | 61.2 | 53.2 | 79.5 | 31.2 | 45.3 | 39.8 | 33.3 | 77.8 | 34.7 | 70.1 | 51.5 | 57.2 | 50.6 | 41.4 | 51.9 | 46.0 | 67.6 | 59.5 | 52.9 |
| SD+DINO (Zhang et al., 2023) | 71.4 | 59.1 | 87.3 | 38.1 | 51.3 | 43.3 | 40.2 | 77.2 | 42.3 | 75.4 | 63.2 | 68.8 | 56.0 | 66.1 | 52.8 | 59.4 | 63.0 | 55.1 | 59.3 |
| **I** NCNet (Rocco et al., 2018) | 17.9 | 12.2 | 32.1 | 11.7 | 29.0 | 19.9 | 16.1 | 39.2 | 9.9 | 23.9 | 18.8 | 15.7 | 17.4 | 15.9 | 14.8 | 9.6 | 24.2 | 31.1 | 20.1 |
| SFNet (Lee et al., 2019) | 26.9 | 17.2 | 45.5 | 14.7 | 38.0 | 22.2 | 16.4 | 55.3 | 13.5 | 33.4 | 27.5 | 17.7 | 20.8 | 21.1 | 16.6 | 15.6 | 32.2 | 35.9 | 26.3 |
| PMD (Li et al., 2021) | 26.2 | 18.5 | 48.6 | 15.3 | 38.0 | 21.7 | 17.3 | 51.6 | 13.7 | 34.3 | 25.4 | 18.0 | 20.0 | 24.9 | 15.7 | 16.3 | 31.4 | 38.1 | 26.5 |
| **K** CATs (Cho et al., 2021) | 52.0 | 34.7 | 72.2 | 34.3 | 49.9 | 57.5 | 43.6 | 66.5 | 24.4 | 63.2 | 56.5 | 52.0 | 42.6 | 41.7 | 43.0 | 33.6 | 72.6 | 58.0 | 49.9 |
| PMNC (Lee et al., 2021) | 54.1 | 35.9 | 74.9 | 36.5 | 42.1 | 48.8 | 40.0 | 72.6 | 21.1 | 67.6 | 58.1 | 50.5 | 40.1 | 54.1 | 43.3 | 35.7 | 74.5 | 59.9 | 50.4 |
| SemiMatch (Kim et al., 2022a) | 53.6 | 37.0 | 74.6 | 32.3 | 47.5 | 57.7 | 42.4 | 67.4 | 23.7 | 64.2 | 57.3 | 51.7 | 43.8 | 40.4 | 45.3 | 33.1 | 74.1 | 65.9 | 50.7 |
| Trans.Mat. (Kim et al., 2022b) | 59.2 | 39.3 | 73.0 | 41.2 | 52.5 | 66.3 | 55.4 | 67.1 | 26.1 | 67.1 | 56.6 | 53.2 | 45.0 | 39.9 | 42.1 | 35.3 | 75.2 | 68.6 | 53.7 |
| SCorrSAN (Huang et al., 2022) | 57.1 | 40.3 | 78.3 | 38.1 | 51.8 | 57.8 | 47.1 | 67.9 | 25.2 | 71.3 | 63.9 | 49.3 | 45.3 | 49.8 | 48.8 | 40.3 | 77.7 | 69.7 | 55.3 |
| SimSC-iBOT (Li et al., 2023) | 62.2 | 54.9 | 79.3 | 53.2 | 57.0 | 72.1 | 64.8 | 77.7 | 39.2 | 75.9 | 69.5 | 68.7 | 62.4 | 59.4 | 45.2 | 49.5 | 86.8 | 71.4 | 63.5 |
| SD4Match-Single | 72.1 | 66.5 | 82.3 | 62.5 | 57.6 | 76.0 | 73.3 | 81.5 | 62.0 | 85.0 | 71.9 | 76.1 | 68.5 | 76.5 | **68.9** | 58.0 | 89.3 | 83.1 | 72.6 |
| SD4Match-Class | 75.1 | 66.6 | **88.1** | **71.4** | 57.8 | **86.6** | 74.6 | **84.2** | 63.0 | 83.8 | 71.5 | 77.6 | **73.5** | **87.2** | 63.3 | 60.0 | **92.0** | **89.8** | **75.5** |
| SD4Match-CPM | **75.3** | **67.4** | 85.7 | 64.7 | **62.9** | **86.6** | **76.5** | 82.6 | **64.8** | **86.7** | **73.0** | **78.9** | 70.9 | 78.3 | 66.8 | **64.8** | 91.5 | 86.6 | **75.5** |

$\mathbf{X} = \{(\mathbf{x}_q^A, \mathbf{x}_q^B) \mid q = 1, 2, ..., n\}$, for each $\mathbf{x}_q^A = (x_q^A, y_q^A)$, we find its predicted correspondence $\bar{\mathbf{x}}_q^B$ and calculate PCK for the image pair by:

$$PCK(I^A, I^B) = \frac{1}{n} \sum_q^n \mathbb{I}(\|\bar{\mathbf{x}}_q^B - \mathbf{x}_q^B\| \leq \alpha * \theta) \qquad (8)$$

where $\theta$ is the base threshold, $\alpha$ is a number less than 1 and $\mathbb{I}(\cdot)$ is the binary indicator function with $\mathbb{I}(\texttt{true}) = 1$ and $\mathbb{I}(\texttt{false}) = 0$. For PF-Pascal, $\theta$ is set as $\theta_{img} = \max(h_{img}, w_{img})$. For PF-Willow, the base threshold is $\theta_{kps} = \max(\max_q(x_q^B) - \min_q(x_q^B), \max_q(y_q^B) - \min_q(y_q^B))$. For SPair-71K, the base threshold is $\theta_{bbox} = \max(h_{bbox}, w_{bbox})$ where $h_{bbox}$ and $w_{bbox}$ are height and width of the bounding box. All three base thresholds' choices align with the literature convention.

## 4.3 EVALUATION RESULTS

**Evaluation on SPair-71k**    We provide the evaluation results on the SPair-71k in Tab. 2. Specifically, we achieve the best results across all 18 categories, and we outperform the second-best method SimSC-iBOT by 12 percentage points (from 63.5 to 75.5) when considering the overall accuracy. Compared to DIFT, which shares the same SD model with ours but uses a textual template as the prompt, SD4Match-Single improves the performance of the SD model by 37.2%, proving that the potential buried in the SD model can be harnessed by simply learning a single prompt. Among the three options of our method, SD4Match-Class outperforms SD4Match-Single by 2.9 percentage points. This echoes the results in Tab. 1, showing the benefit of the prior knowledge of the object. SD4Match-CPM achieves the same accuracy as SD4Match-Class, indicating the effectiveness of our CPM module in capturing the prior knowledge of the object without manual effort.

**Generalizability Test**    Following the practice in the literature, we also test the generalizability of our method by tuning the prompt on the training data of PF-Pascal and evaluating it on the testing data of PF-Pascal, PF-Willow, and SPair-71k. The results are presented in Tab. 3. We do not evaluate SD4Match-Class since the three datasets have different categories. Among methods using image-level annotations and keypoint annotations, SD4Match-CPM achieves accuracy on par with SimSC-iBOT on PF-Pascal, and the best generalized results on PF-Willow and SPair-71k across all $\alpha$. This verifies the generalizability of our method. Compared with zero-shot methods, especially SD-based methods DIFT and SD+DINO, we observe substantial improvement on PF-Pascal and PF-Willow but deterioration on SPair-71k. This is because our universal prompt and CPM overfit the smaller distributions of PF-Pascal and PF-Willow, leading to a certain degree of reduced generalizability on the much larger distribution of SPair-71k. Other methods tuned on PF-Pascal also exhibit this trend. To further address this point, we additionally provide the results of our method tuned on SPair-71k. We observe a slight improvement on PF-Pascal but a substantial boost on PF-Willow when compared

Table 3: Generalizability test of different methods. All methods are either zero-shot or trained on the PF-Pascal dataset unless labelled otherwise.

| | Method | PF-Pascal $\theta_{img}$ @ $\alpha$ | | | PF-Willow $\theta_{kps}$ @ $\alpha$ | | | SPair-71k $\theta_{bbox}$ @ $\alpha$ | |
|---|---|---|---|---|---|---|---|---|---|
| | | 0.05 | 0.1 | 0.15 | 0.05 | 0.1 | 0.15 | 0.05 | 0.1 |
| **Z** | DINOv1 (Caron et al., 2021) | 55.6 | 74.2 | 81.6 | 39.7 | 64.8 | 77.0 | 23.1 | 35.9 |
| | DINOv2 (Oquab et al., 2023) | 63.4 | 82.6 | 89.9 | 37.3 | 63.4 | 73.1 | 38.4 | 53.9 |
| | DIFT (Tang et al., 2023) | 69.0 | 82.2 | 88.1 | 44.8 | 68.0 | 79.8 | 39.7 | 52.9 |
| | SD+DINO (Zhang et al., 2023) | 68.1 | 85.7 | 91.5 | 44.8 | 72.0 | 85.0 | 44.1 | 59.3 |
| **I** | NCNet (Rocco et al., 2018) | 54.3 | 78.9 | 86.0 | 44.0 | 72.7 | 85.4 | - | 26.4 |
| | SFNet (Lee et al., 2019) | 59.0 | 84.0 | 92.0 | 46.3 | 74.0 | 84.2 | 11.2 | 24.0 |
| | PWarpC-NCNet (Truong et al., 2022) | 64.2 | 84.4 | 90.5 | 45.0 | 75.9 | 87.9 | 18.2 | 35.3 |
| **K** | CATs (Cho et al., 2021) | 75.4 | 92.6 | 96.4 | 40.9 | 69.5 | 83.2 | 13.6 | 27.0 |
| | Trans.Mat (Kim et al., 2022b). | 80.8 | 91.8 | - | - | 65.3 | - | - | 30.1 |
| | DHPF (Min et al., 2020) | 77.3 | 91.7 | 95.5 | 44.8 | 70.6 | 83.2 | 15.3 | 27.5 |
| | SimSC-iBOT (Li et al., 2023) | **88.4** | **95.6** | 97.3 | 44.9 | 71.4 | 84.5 | 22.0 | 37.9 |
| | SD4Match-Single | 81.3 | 92.4 | 96.6 | 50.7 | 77.8 | 89.5 | **30.5** | **44.4** |
| | SD4Match-CPM | 84.4 | 95.2 | **97.5** | **52.1** | **80.4** | **91.2** | 27.2 | 40.9 |
| | SD4Match-Single (Tuned on SPair-71k) | 71.8 | 85.5 | 90.4 | 55.5 | 81.3 | 91.2 | 56.5 | 72.6 |
| | SD4Match-CPM (Tuned on SPair-71k) | 73.3 | 87.0 | 91.5 | 56.7 | 80.9 | 91.6 | 59.5 | 75.5 |

| CPM conditioned on: | SPair-71k @ $\alpha = 0.1$ |
|---|---|
| 1. Image Pair; Local Feat. | 75.5 |
| 2. Image Pair; Global Desc. | 73.3 |
| 3. Ind. Image; Local Feat. | 70.8 |
| 4. Ind. Image; Global Desc. | 68.5 |
| 5. w/o global prompt | 74.6 |
| 6. w/o $g_n(\cdot)$ | 74.0 |

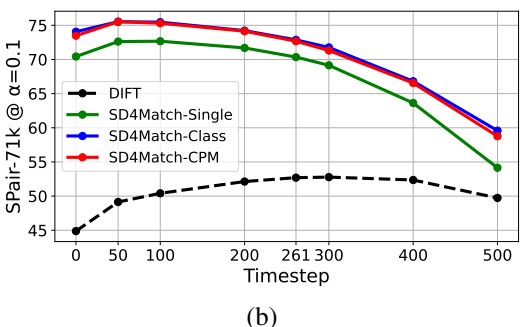

(a)

(b)

Figure 3: Results of ablation studies. (a): Evaluation of our method on SPair-71k with different settings of CPM. (b): Evaluation of our method on SPair-71k at different timesteps.

to zero-shot baselines. This improvement is attributed to SD4Match being tuned on a larger dataset, which prevents overfitting on a small data distribution and enhances its generalizability.

## 4.4 ABLATION STUDIES

**Conditional Prompting Module** We conduct a thorough ablation study to evaluate each design choice of CPM and present the results in Fig. 3 (a). We evaluate our choice of conditioning in cases 1 through 4. Case 1 involves conditioning on the image pair and local feature patches, representing our current setting in CPM. When we replace local feature patches with a global image descriptor (case 2) or shift from conditioning on the image pair to conditioning on an individual image (case 3), there is a decline in performance for both scenarios. Notably, the performance drop from case 1 to case 3 is more significant than that from case 1 to case 2. This suggests that conditioning on the image pair has a more substantial impact than conditioning on local feature patches in enhancing matching. In Case 4, where we condition on the individual image and its global image descriptor, the architecture mirrors the implicit captioner in SD+DINO and COCOOP (Zhou et al., 2022a). This configuration results in the lowest accuracy, further emphasizing the advantages of conditioning on an image pair and leveraging local features in semantic matching. We also investigate the impacts of the global prompt and the patch-wise linear layer $g_n(\cdot)$ in cases 5 and 6, respectively. Removing either of these elements leads to a decrease in performance, underscoring their effectiveness.

**Evaluation at Different Timesteps** The timestep $t$ is an important hyperparameter that plays a major role in the matching quality. Both DIFT and SD-DINO have performed the grid search to find the optimal timestep. We test our method using different $t$ and plot the result in Fig. 3 (b). We

| Bottle Samples | Bottle Generation | TV Samples | TV Generation | Car Samples | Car Generation |

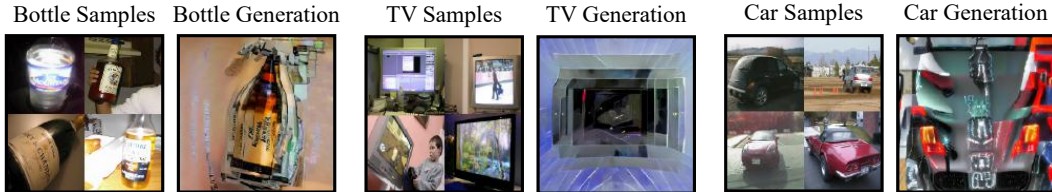

Figure 4: Visualization of the learned class-specific prompt in SD4Match-Class.

| Image A | Image B | Generated Image | Image A | Image B | Generated Image |

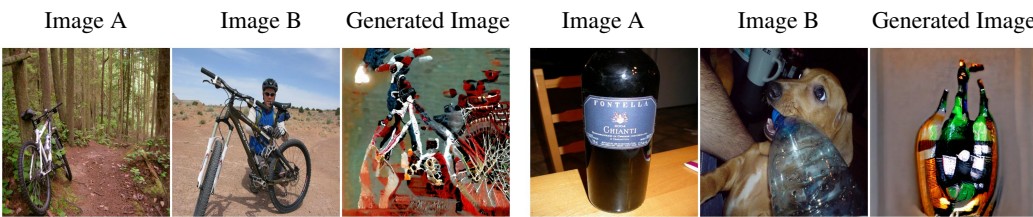

Figure 5: Visualization of the learned conditional prompt in SD4Match-CPM.

select DIFT as the baseline because we share the same SD model and do not involve other types of feature. As illustrated, our method outperforms the baseline across a wide range of $t$ over 0 to 500. Interestingly, we train our method at $t = 261$ which is the optimal value under the zero-shot setting, but the accuracy at inference instead peaks at $t$=50 and then gradually decreases. This indicates that the model favors a cleaner input image (fewer $t$) but the noise is also necessary to achieve a good result when using the learned prompt.

**Visualization of SD4Match Prompt**   To further investigate what has been learned during prompt tuning, we visualize images generated by Stable Diffusion using the learned prompt. We first visualize the images generated using the class-specific prompt learned by SD4Match-Class and present samples of selected categories in Fig. 4. Interestingly, for each of the selected categories, the generated image is an abstract illustration of that category. This highlights the intriguing capability of the prompt to learn high-level category details using only keypoint supervision at the UNet's intermediate stage. This can be loosely compared with textual inversion (Gal et al., 2022) or DreamBooth (Ruiz et al., 2023), which extract an object's information from multiple images of itself and generate the same object in different styles. We show that, even without the explicit reconstruction supervision present in these works, Stable Diffusion can still learn the category-level information from local-level supervision. This reveals the powerful inference ability of the SD model on local information. We also visualize the conditional prompt generated by the CPM and provide selected samples in Fig. 5. The conditional prompt, similar to the class-specific prompt, captures the semantic information of objects' categories. Moreover, the prompt emphasizes the shared object between two images. As shown in Fig. 5, multiple objects are present in the image pair, and the prompt focuses on the object with the same semantic meaning. This suggests that the CPM is effective in automatically capturing the prior knowledge of the object of interest, subsequently enhancing the matching accuracy.

## 5   CONCLUSION

In this paper, we introduce SD4Match, a prompt tuning method that adapts the Stable Diffusion for the semantic matching task. We demonstrate that the quality of features produced by the SD model for this task can be substantially enhanced by simply learning a single universal prompt. Furthermore, we present a novel conditional prompting module that conditions the prompt on the local features of an image pair, resulting in a notable increase in matching accuracy. We evaluate our method on three public datasets, establishing new benchmark accuracies for each. Notably, we surpass the previous state-of-the-art on the challenging SPair-71k dataset by a margin of 12 percentage points.

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

## A  MORE VISUALIZATIONS OF THE LEARNED PROMPT

We provide more visualizations of images generated by class-specific prompts and conditional prompts in  Fig. 6 and  Fig. 7 respectively.

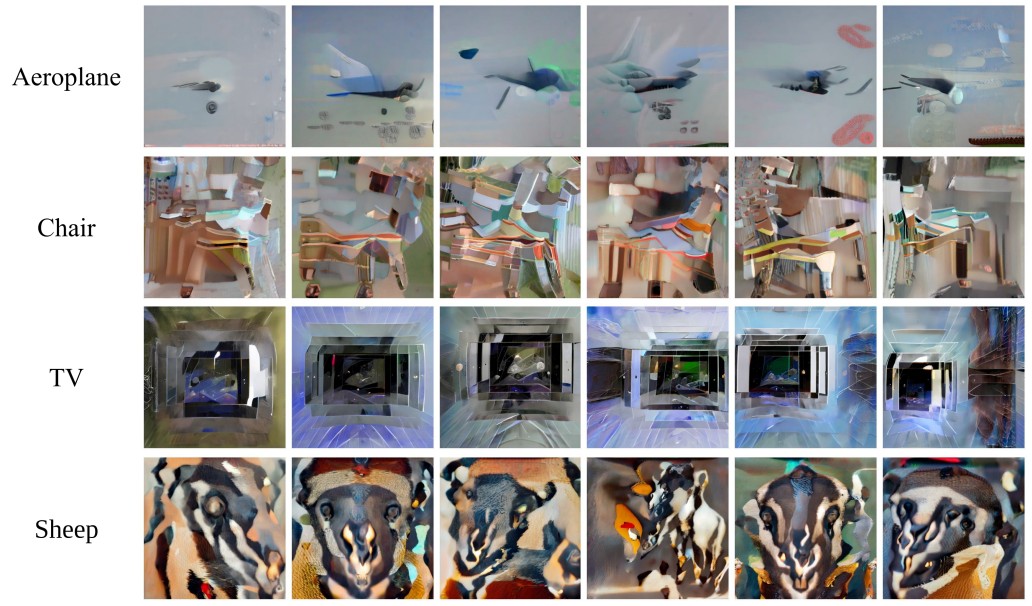

Figure 6: Visualizations of images generated by class-specific prompts learned by SD4Match-Class

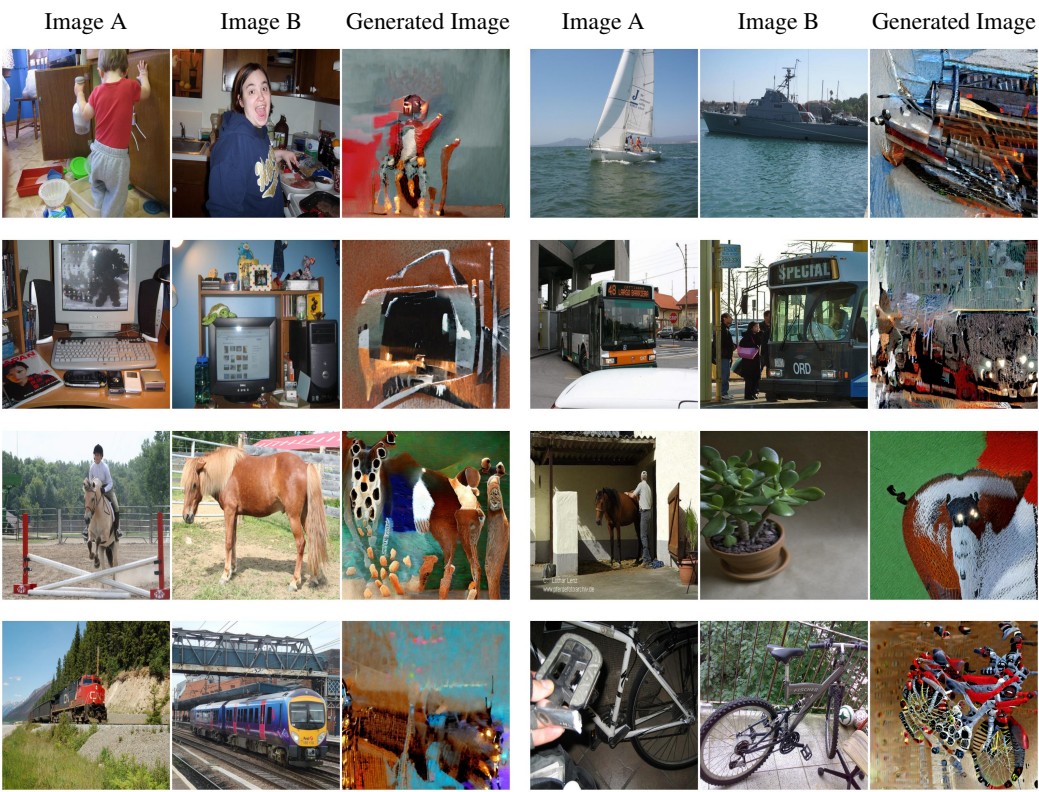

Figure 7: Visualization of images generated by conditional prompts learned by SD4Match-Class.

# B QUALITATIVE MATCHING RESULTS

We provide qualitative comparisons in matching accuracy between DIFT, SD+DINO, and SD4Match-CPM.

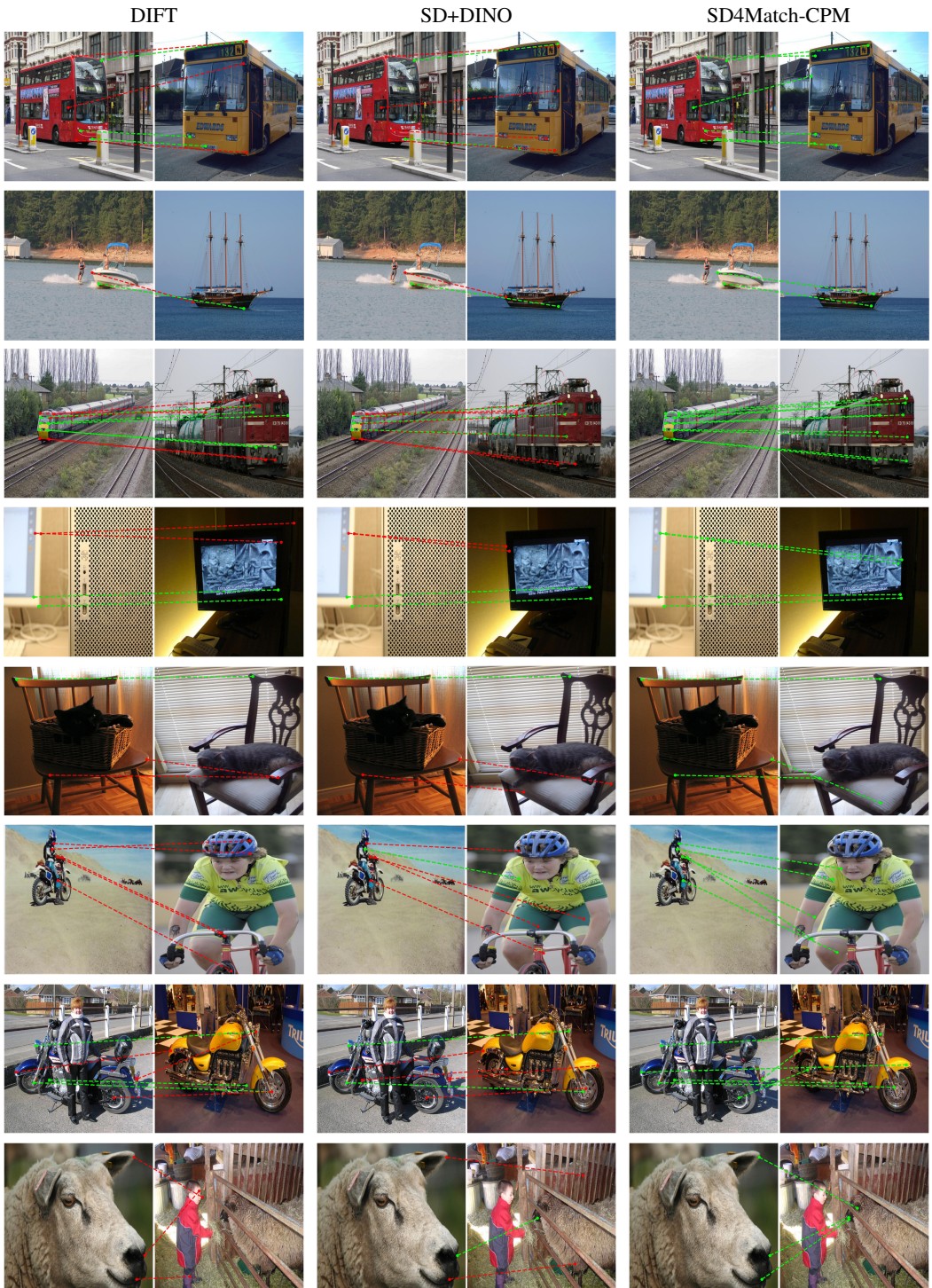

Figure 8: Qualitative comparison between DIFT, SD+DINO, and SD4Match-CPM.

