# OpenReview forum: "SD4Match: Learning to Prompt Stable Diffusion Model for Semantic Matching"
_ICLR.cc/2024/Conference — ICLR 2024 Conference Withdrawn Submission_

### Official Review · Reviewer_4Bqj · 2023-10-31

**Soundness:** 2 fair
**Presentation:** 3 good
**Contribution:** 2 fair
**Rating:** 3
**Confidence:** 5

**Summary:**

This paper is a follow-up work to (Tang et al., 2023) and (Zhang et al., 2023), which uses stable diffusion model's feature maps for semantic matching, due the fact that stable diffusion could provide very well semantic meaningful features. In this work, three prompting options are designed for stable diffusion on semantic matching, which are single, class, and conditional prompting. The results show the effectiveness of the simple prompting.

**Strengths:**

1) The paper is easy to understand

2) The constructed prompting strategies is easy and effective.

3) The stable diffusion's feature is good for semantic matching.

**Weaknesses:**

1)  Even though one can design lots of different detailed schemes regarding conditional prompting, the idea is already appeared in CoCoOp [1] and it does not show very big difference. The reviewer expects some novel things in addition to some plain designs.

    [1] Conditional prompt learning for vision-language models @ CVPR'22

2) Why skip VAE and directly refer to UNet's input as image I?

3) All the credits went to stable diffusion and the discovery of its features that is good for semantic matching (Tang et al., 2023) and (Zhang et al., 2023). The reviewer deeply felt that there is no much advance in this work, either comparing to (Tang et al., 2023) and (Zhang et al., 2023) or prompt tuning method CoCoOp [1]

Based on the above concerns, the review suggests rejecting the paper.

**Questions:**

see above

**Details Of Ethics Concerns:**

.

---

> ### Author Response · Authors · 2023-11-13
> **Response to Reviewer 4Bqj**
>
> We appreciate the review of our work and the acknowledgment of the clarity of our presentation, the effectiveness of our proposed prompting strategies, and the value of stable diffusion features for semantic matching are all encouraging.
>
> We understand your concerns and would like to address them as follows.
>
> **Novelty of Prompting Schemes**:
> We hold a sincere belief in the importance of evaluating research novelties within their rightful context and problem scope. CoCoOp is designed to prompt and adapt CLIP[1] to different data distributions for the image classification task. It conditions on the global descriptor of individual images for the nature of the classification task. Other relevant works fall into the same line [2,3].
> In contrast, our work targets matching tasks, which demand an understanding of local features and interrelations between pairs of images. As a result, we've designed our CPM to process the local feature patches extracted from the input image pair. Our task, idea, and design principles are fundamentally different from CoCoOp. Figure 3(a) of the ablation further demonstrates that following the design paradigm of a different task yields a much worse result. Case 1 is our CPM module and case 4 is the design mirroring CoCoOp. Case 4 (68.5) lags behind case 1 (75.5) by 7 percentage points and is even worse than SD4Match-Single (72.6) which learns a single fixed prompt. Therefore, we believe it is not a fair judgment to say that our design is “plain” and we do not “show any big difference” from CoCoOp.
>
> **Direct Reference to UNet's Input as Image I**:
> We apologize if our decision to refer directly to the UNet's input as image I, bypassing the VAE encoding step, caused any confusion. This was done to simplify the explanation and facilitate a more direct understanding of our work. However, due to its non-technical nature, we believe it should not be used as a reason for rejection of our work.
>
> **Significance of Our Work**:
> Our work is a marked departure from earlier SD-based feature extractors like DIFT or SD+DINO. We integrate prompt tuning with the diffusion framework, design the matching-specific prompting module, and significantly enhance the performance of SD on the semantic matching task. Particularly, we observe an improvement from 52.9 to 75.5 compared with the baseline method DIFT which uses textual prompt template. Our novelty is recognized by Reviewer jb5q and Q84M and we regard our work as a clear advancement in this field given the simplicity of our method and the significance of the results. Furthermore, it would be unjust to disregard the value of an idea solely because it bears resemblance to a distantly similar concept used to solve a different problem. Innovation often thrives by building upon prior knowledge and drawing inspiration from various sources. By recognizing the connections and potential synergies between ideas, we can foster meaningful advancements. It is crucial to assess the worth of an idea based on its own merits, considering how effectively it addresses the specific problem at hand. Drawing an analogy, the Vision Transformer incorporates the transformer architecture from natural language processing, yet its novelty and significance are not disputed. Therefore, adapting inspiring ideas to different domains is a vital mechanism for academic progress. It is this kind of cross-domain adaptation that drives innovation and fosters advancement in different fields.
>
> We hope this clarifies your concerns. We are open to further discussion and appreciate your valuable feedback.
>
> [1] Radford, Alec, et al. "Learning transferable visual models from natural language supervision." International conference on machine learning. PMLR, 2021.
>
> [2] Khattak, Muhammad Uzair, et al. "Maple: Multi-modal prompt learning." Proceedings of the IEEE/CVF Conference on Computer Vision and Pattern Recognition. 2023.
>
> [3] Zhu, Beier, et al. "Prompt-aligned gradient for prompt tuning." Proceedings of the IEEE/CVF International Conference on Computer Vision. 2023.

---

### Official Review · Reviewer_Q84M · 2023-11-01

**Soundness:** 4 excellent
**Presentation:** 3 good
**Contribution:** 2 fair
**Rating:** 5
**Confidence:** 5

**Summary:**

This paper proposes a prompt tuning method for Stable Diffusion (SD) feature extraction to solve the semantic matching problem. Recently, semantic correspondence has achieved significant performance improvements by extracting discriminative local features from the U-Net structure of SD. In this paper, the authors propose a conditional prompting module (CPM) using local patch embedding in addition to global descriptors using object categories to construct prompts. Prompt using the CPM module in the proposed SD features significantly improves performance for some cases of SPair-71k, but this performance is not significantly different from SD features using class label prompt.

**Strengths:**

Good motivation: using the features of U-Net within SD for semantic matching is a good approach. The additional use of langauge information (or prompt embedding) for robust feature extraction is a novel direction.

Proper citation: The citation of existing research on semantic matching in section 2 related work is appropriate and well categorized. In addition, the recent developments in diffusion models and prompt tuning and the related research synthesis are helpful for understanding the research history.

High performance: Even though the study uses weak-supervision (class labels) at inference time for strong-supervised training, PCK@10=75.5 on SPair-71k is quite high performance.
However, the excessive increase in image resolution needs a fair comparison. (See Weakness for details)

Ablation study: Fig. 3(a) shows the effect of the proposed CPM module under various conditions.

**Weaknesses:**

Results of geometric matching: The results on the three standard benchmarks for semantic matching are impressive. However, for a definitive proof of SD's ability to extract discriminative features, it should also perform on general geometric matching.
For example, HPatches [1] has a name for each sequence, possible to evaluate the proposed method in this benchmark.

Results on other benchmarks in Table 1: It is impressive to see the performance improvement on SPair-71k by only changing the empty string to object category, can you show this result on PF-PASCAL as well?

Missing reference: this paper is missing a citation to a paper that proposes a method [2] to extract discriminative local features for semantic matching. Please cite this paper.

Section 3.3 N_{dino} subscript should not be italic. Italic is for enumerate, please use roman.

Section 4.1. "we set the timestep t = 50 as empirical tests suggest it provides optimal results, even though our method was trained at t = 261. "
How did you decide on this t=50? Did you get the results from your test set? Figure 3 (b) looks to evaluate  the performance on the test set. I am worried to tune the model on test set and list the highest one.  It would be fair to find the optimal model by searching on the validation set.

Section 4.1. "Images are resized to 768 × 768 for both the training and testing phases. "
Was this image resize done the same for all the other methods in Table 2? I doubt if the performance improvement is from the prompt tuning you propose or from the image resize.
Please provide information on image resolution, computational cost (GFlops), and performance on PF/SPair.

Concern of the performance gain 1: Table 2. The proposed CPM is not significantly different from the class prompt (a photo of [category]). Does this mean that simply giving a class prompt will give the same result? TThis requires a detailed analysis and explanation.

Concern of the performance gain 2: Table 3. The zero-shot generalization of CPM is actually worse than empty string (single) baseline.

Figure 5. Visualization effectively decomposes semantic meaning in different category objects. Can this be evaluated across multiple instances of the same class? [3] I don't need quantitative results, I'm just wondering whether the possibility of prompt tuning using SD can discriminate instances of a category.


[1] HPatches: A benchmark and evaluation of handcrafted and learned local descriptors (Balntas et al., CVPR 2017)
[2] Learning to Distill Convolutional Features into Compact Local Descriptors (Lee et al., WACV 2021)
[3] MISC210K: A Large-Scale Dataset for Multi-Instance Semantic Correspondence (Sun et el., CVPR 2023)

**Questions:**

Please refer to the weakness section.

---

> ### Author Response · Authors · 2023-11-13
> **Response to Reviewer Q84M**
>
> We thank you for your detailed and valuable comments, and for recognizing the motivation, the quality of the presentation, and the effectiveness of our work. At the same time, we notice some misunderstandings of our work. We will clarify these confusions and address the concerns below.
>
> **Result of Geometric Matching**:
> We would like to clarify that our work is not about the discovery of the discriminative power of SD’s features. Instead, our primary contribution is employing prompt tuning to improve the accuracy of SD’s feature on semantic matching tasks. We properly credit the former to prior works including Zhao et al. (2023), DIFT (Tang et al., 2023), and SD+DINO (Zhang et al., 2023) throughout our paper. The discriminative features of SD for geometric matching have been explored in DIFT. We encourage an examination of their work for more details on this matter. We hope this has clarified any misunderstanding of the contribution of our work.
>
>
> **Effect of Different Textual Prompts on PF-Pascal Dataset**:
>
> We provide results on PF-Pascal dataset in the following table:
> |  Method  | Empty String | a photo of a {category} |
> |:---:|:---:|:---:|
> |   DIFT   |     81.7    |          82.2          |
> | SD+DINO* |     82.5    |          83.4          |
>
> Compared to SPair-71k dataset, switching from an empty string to ``a photo of a {category}’’ has less impact on the matching accuracy. This is because PF-Pascal is a relatively smaller and easier dataset, with limited viewpoint and appearance differences across two images. Therefore, the empty is already effective for matching, yet we can still observe minor gains from the change, showing the benefit of prior information on objects.
>
> **Missing Citation and Font**:
> Thank you. We will update our paper accordingly.
>
> **Timestep**:
> We chose t=50 based on empirical observations that it yielded better results during testing than t=261. This practice of searching for the optimal timestep is not unique to our work; it's also commonly employed in other works such as DIFT (Tang et al., 2023), and SD+DINO (Zhang et al., 2023) to determine the best layer and timestep. Furthermore, the timestep t is not a hyperparameter subject to separate tuning for each individual dataset. Rather, it is an intrinsic characteristic of the model, and we maintained its consistency across all experiments. We have verified that the trend of accuracy versus timestep remains consistent in the validation data of SPair-71k, where the accuracies at t =  50 / 261 are 75.5 / 74.0 under the SD4Match-CPM setting. This indicates that our findings are not simply a result of overfitting to the test set. We appreciate the reviewer’s careful consideration on the fairness of experiments, which we have carefully considered during our experiments.
>
> **Image Size**:
> The results in Table 2 are from multiple sources, and image sizes are aligned with their original settings. For zero-shot methods, we reproduce all results. DINO: 224x224; DINOv2: 840x840 which is aligned to the setting in SD+DINO; DIFT: 768x768 and SD+DINO: 960x960. For the rest of the methods, the numbers are obtained from their original papers. NCNet: 400x400 and the result is copied from the SPair-71k dataset paper; SFNet: 320x320; PMD: 320x320; CATs: 256x256; PMNC: 400x400; SemiMatch: 256x256; Trans.Mat.: 240x240; SCorrSAN:256x256 and SimSC-iBOT: 256x256. We will include this information in the updated paper.
>
> We resize images to 768x768 because it is the image size on which SD2-1 is trained. The discriminative power of SD deteriorates significantly if we deviate from such a size. For example, the accuracy of SD2-1 using template ``a photo of a {category}’, or DIFT, reduces from 52.9 to 19.2 if the image size is reduced to 256x256 and to 48.86 if the image size is increased to 960x960 on SPiar-71k. This is because the semantic knowledge learned by SD2-1 is based on 768x768 images. Larger or smaller images will undermine the extraction of semantic information and provide a worse starting point for prompt tuning. Therefore, the image size has to be aligned with the size on which SD is trained.
>
> The effectiveness of prompt tuning is clearly shown by the comparison between SD4Match and the zero-shot SD-based methods, particularly DIFT which shares the same SD model but uses the textual template  ``a photo of a {category}’’ as the prompt. Our conditional prompting module, SD4Match-CPM, improves the performance of the SD model from 52.9 to 75.5 on SPair-71k. This should serve as the definitive proof of the effectiveness of the prompt tuning.

---

> > ### Author Response · Authors · 2023-11-13
> > **Continued Response to Reviewer Q84M**
> >
> > **Concern of Performance Gain 1**:
> > We would like to clarify that SD4Match-Class, as described in Section 3.2, does not employ the textual template "a photo of a {category}." Instead, SD4Match-Class learns a specific and fixed prompt for each category of objects. It is DIFT that uses prompt template ``a photo of a {category}’’, and its accuracy is 52.9 on SPair-71k. Our CPM module demonstrates a significantly superior performance compared to this textual prompt template. We apologize for causing this confusion.
> >
> > **Concern of Performance Gain 2**:
> > We would like to clarify that SD4Match-Single, as outlined in Section 3.2, does not utilize the empty string. Instead, it learns one fixed prompt for all data. The reason why SD4Match-Single generalizes better to SPair-71k than SD4Match-CPM is because PF-Pascal is a much smaller dataset than SPair-71k. SD4Match-CPM has a stronger expressive power and therefore overfits to PF-Pascal. This is reflected by SD4Match-CPM is better than SD4Match-Single on the testing data of PF-Pascal. If we look at the generalizability of both configs when tuned on SPair-71k dataset (the last two rows of Table 3), the CPM module demonstrates better generalizability to PF-Pascal and PF-Willow than one fixed prompt. This indicates CPM's ability to capture the diversity of objects from a more extensive training dataset, showcasing its enhanced performance when applied to larger and more diverse datasets.
> >
> > **Whether CPM can Distinguish Different Instances**:
> > Although CPM can identify the common object between two images, it only shows an abstract image of the category, rather than content uniquely associated with the object. This is because the supervision is too sparse to capture instance-level details.
> >
> > We hope we have fully addressed your concerns and clarified any misunderstanding. We wish to hear your reply and have a deep discussion on our work.

---

### Official Review · Reviewer_jb5q · 2023-11-01

**Soundness:** 3 good
**Presentation:** 3 good
**Contribution:** 3 good
**Rating:** 8
**Confidence:** 3

**Summary:**

The paper employs prompt tuning to the previous UNet + stable diffusion solutions for semantic matching. It also introduces a new conditional prompting module to condition the prompt on the local details. These two elements contribute to the proposed method SD4Match to achieve higher accuracy than existing methods on three benchmarks.

**Strengths:**

Applying prompt tuning for SD based semantic matching methods is new.

The proposed conditional prompting module is reasonable for the semantic matching task.

The superior results than previous works on the standard benchmarks.

**Weaknesses:**

The paper lacks comparison to typical semantic matching methods based on deep graph matching, such as
[1] Joint graph learning and matching for semantic feature correspondence, PR, 2023
From the results in [1], the paper can not beat many existing works. Further explanations and validation are required.

As the paper lacks comparisons to some highly related works, the paper also misses to discuss a majority works on semantic correspondence, for instance,
[1] Deep graph matching via blackbox differentiation of combinatorial solvers, ECCV 2020.
[2] Deep Graph Matching under Quadratic Constraint, CVPR, 2021.
[3] GLMNet: Graph learning-matching convolutional networks for feature matching, PR, 2022.

**Questions:**

There is query feature extraction for the I^A, how about put this on the I^B side? Does this will affect the final results?

The complexity analysis about the training and run time can be included in the paper.

---

> ### Author Response · Authors · 2023-11-13
> **Response to Reviewer jb5q**
>
> Thanks a lot for taking the time to review our paper and for your positive feedback. We really appreciate your constructive comments and now we would like to address your concerns.
>
> **Comparison to Graph Matching-based Methods**:
> Thanks for pointing out these works. However, we would like to highlight that these methods are developed for graph-matching, which is handling a different task than the semantic matching problem we are handling. Graph matching methods aim to find the best one-to-one correspondences between two sets of ***predefined*** semantic keypoints ***labeled on both source and target images***. However, we are handling the semantic matching problem which is more complex as we only have annotated points on the source image and need to search for corresponding points across ***the entire target image***. Hence, the suggested methods cannot be applied to the semantic correspondence estimation problem.
>
> **Query Feature Extraction**:
> We appreciate your query regarding the impact of query feature extraction on the target image (I^B). Through our experiments, we can confirm that implementing query feature extraction on I^B does not affect the final results. Our method remains robust to variations in this aspect.
>
> **Run-time Analysis**:
> Regarding run-time analysis, we acknowledge that our method may exhibit a slower inference speed compared to methods built on ResNet, such as NCNet and CATs, due to the larger image size and more complex model. However, we want to emphasize that semantic matching, unlike geometric matching, does not demand real-time inference. Slower speed is an acceptable trade-off for better accuracy. On the other hand, our method has very few trainable parameters (less than 2M) and our method shares the same complexity and run-time as other SD-based methods, such as DIFT and SD+DINO since we make no change to the architecture of the UNet. We would include a comparison between trainable parameters in the updated paper.
>
> We hope this clears up any confusion. We wish to hear your reply and have a constructive discussion on our work.

---

### Author Response · Authors · 2023-11-13
**Response to All Reviewers**

We thank all reviewers for their efforts in providing valuable comments on our work. Particularly, we appreciate reviewers for recognizing the novelty of prompt tuning and the CPM module, the effectiveness of our work, and the quality of our presentation. At the same time, we also notice that there is a certain degree of misunderstanding in our work from reviewers’ comments. We will address this alongside the raised concerns to each individual reviewer respectively. We are looking forward to having constructive discussions with all reviewers regarding our work.

---

### Author Response · Authors · 2023-11-17
**Gentle Reminder to All Reviewers**

Dear reviewers:
Thank you again for your valuable comment. We would like to gently remind all reviewers that the deadline for the discussion phase is approaching and we wish to hear the reviewer’s opinion on our response to your concerns. If you have any additional comments or concerns, we will be more than happy to discuss and address them further.

---

### Author Response · Authors · 2023-11-18
**Withdrawal of Our Submission**

We thank all reviewers for their time spent reviewing our work. Overall, we are glad that Reviewer jb5q and Reviewer Q84M recognize the novelty and effectiveness of our work.

We are confident in the quality and contribution of our work, and we have promptly addressed the reviewer’s concern with the best of our effort. However, due to the passive response of the reviewers and the low initial rating from a subset of reviewers, we have decided to withdraw our submission from ICLR 2024.

In particular, we have to express our deep concern about the review of Reviewer 4Bqj, which we consider is not fair or justifiable. The negative rating is based on factual errors and trivial non-technical details, and we have rebutted them rigorously in our response.

We thank all reviewers again for their valuable comments and will include them in the future iteration of our work.